# Preoperative Chemoradiotherapy for Gastroesophageal Junction Adenocarcinoma Modified by PET/CT: Results of Virtual Planning Study

**DOI:** 10.3390/medicina57121334

**Published:** 2021-12-06

**Authors:** Marek Slavik, Petr Burkon, Iveta Selingerova, Pavel Krupa, Tomas Kazda, Jaroslava Stankova, Tomas Nikl, Renata Hejnova, Zdenek Rehak, Pavel Osmera, Tomas Prochazka, Eva Dvorakova, Petr Pospisil, Peter Grell, Pavel Slampa, Radka Obermannova

**Affiliations:** 1Department of Radiation Oncology, Masaryk Memorial Cancer Institute, Zluty Kopec 7, 656 57 Brno, Czech Republic; slavik@mou.cz (M.S.); krupa@mou.cz (P.K.); tomas.kazda@mou.cz (T.K.); jaroslava.stankova@mou.cz (J.S.); tomas.nikl@mou.cz (T.N.); tprochazka@mou.cz (T.P.); eva2.dvorakova@mou.cz (E.D.); ppospisil@mou.cz (P.P.); slampa@mou.cz (P.S.); 2Department of Radiation Oncology, Faculty of Medicine, Masaryk University, Kamenice 753/5, 625 00 Brno, Czech Republic; 3Research Centre for Applied Molecular Oncology (RECAMO), Masaryk Memorial Cancer Institute, Zluty Kopec 7, 656 53 Brno, Czech Republic; iveta.selingerova@mou.cz; 4Department of Pharmacology, Faculty of Medicine, Masaryk University, Kamenice 753/5, 625 00 Brno, Czech Republic; 222873@mail.muni.cz; 5Department of Nuclear Medicine, Masaryk Memorial Cancer Institute, Zluty Kopec 7, 656 53 Brno, Czech Republic; rehak@mou.cz (Z.R.); pavel.osmera@mou.cz (P.O.); 6Department of Comprehensive Cancer Care, Faculty of Medicine, Masaryk University, Kamenice 735/5, 625 00 Brno, Czech Republic; grell@mou.cz (P.G.); obermannova@mou.cz (R.O.); 7Department of Comprehensive Cancer Care, Masaryk Memorial Cancer Institute, Zluty Kopec 7, 656 53 Brno, Czech Republic

**Keywords:** gastroesophageal junction cancer, PET/CT, radiotherapy, neoadjuvant chemoradiotherapy

## Abstract

*Background and Objectives*: The treatment of gastroesophageal junction (GEJ) adenocarcinoma consists of either perioperative chemotherapy or preoperative chemoradiotherapy. Radiotherapy (RT) in the neoadjuvant setting is associated with a higher probability of resections with negative margins (R0) and better tumor regression rate, which might be enhanced by incrementing RT dose with potential impact on treatment results. This virtual planning study demonstrates the feasibility of increasing the dose to GEJ tumor and involved nodes using PET/CT imaging. *Materials and Methods*: 16 patients from the chemoradiotherapy arm of the phase II GastroPET study were treated by a prescribed dose of 45.0 Gray (Gy) in 25 fractions. PET/CT was performed before treatment. The prescribed dose was virtually boosted on PET/CT-positive areas to 54.0 Gy by 9 Gy in 5 fractions. Dose-volume histograms (DVH) were compared, and normal tissue complication (NTCP) modeling was performed for both dose schedules. *Results*: DVHs were exceeded in mean heart dose in one case for 45.0 Gy and two cases for 54.0 Gy, peritoneal space volume criterion V_45Gy_ < 195 ccm in three cases for 54.0 Gy and V_15Gy_ < 825 ccm in one case for both dose schedules. The left lung volume of 25 Gy isodose exceeded 10% in most cases for both schedules. The NTCP values for the heart, spine, liver, kidneys and intestines were zero for both schemes. An increase in NTCP value was for lungs (median 3.15% vs. 4.05% for 25 × 1.8 Gy and 25 + 5 × 1.8 Gy, respectively, *p* = 0.013) and peritoneal space (median values for 25 × 1.8 Gy and 25 + 5 × 1.8 Gy were 3.3% and 14.25%, respectively, *p* < 0.001). *Conclusion:* Boosting PET/CT-positive areas in RT of GEJ tumors is feasible, but prospective trials are needed.

## 1. Introduction

The role of radiation therapy (RT) in the management of gastroesophageal junction (GEJ) adenocarcinoma is not clearly established [1,2,3,4]. In the western world, the standard of care is either perioperative chemotherapy (POC) or preoperative chemoradiotherapy (PCRT) [2,5]. In the preoperative setting, the CROSS trial showed a significant survival advantage and higher pathological response rate in the arm of concurrent chemoradiotherapy compared to the surgery alone, and the better local control and longer follow-up showed reduced locoregional recurrences in the concurrent chemoradiotherapy arm, and to a lesser extent reduced systemic recurrences [6]. Additionally, the preliminary results of the recently published NeoAgis trial comparing perioperative chemotherapy with ECF/ECX (epirubicin, cisplatin (oxaliplatin), 5-FU (capecitabine)) and more latterly FLOT (docetaxel, 5-FU, leucovorin, oxaliplatin) to preoperative CROSSchemoradiotherapy (carboplatin/paclitaxel, 41.4 Gray (Gy) radiation therapy) showed a higher rate of resection with negative margins (R0, 95% vs. 82%) and better tumor regression rate (TRG ≥ 2 41.7% vs. 12.1%) and the number of local controls was higher in the chemoradiotherapy arm [7]. Also, the TOPGEAR trial comparing POC versus PCRT with subsequent postoperative chemotherapy in GEJ or gastric cancer (GC) demonstrated the safety of administration of preoperative chemoradiotherapy with no added perioperative toxicity. Nevertheless, definitive results are pending [8]. Moreover, an achievement of significant TRG and pathological complete tumor regression (pT0) resection seems to be associated with better overall survival [9,10]. The evidence of significantly improved prognosis in patients reaching complete pathological response after neoadjuvant radiochemotherapy [11] and the fact that higher dose usually means a higher probability of reaching the complete remission of the disease [12,13] led us to conduct this planning study evaluating the safety of possibly boosting the primary tumor and involved nodes in GEJ cancers with the hypothetical consequence of a higher pathological complete response rate with the same RT-related toxicity. The aim of this virtual planning study was the objective feasibility and safety of increased-dose RT. For this purpose, additional virtual boost plans with an increased dose of 9.0 Gy in 5 fractions on tumor and involved nodes using PET/CT imaging were created and added to existing and delivered basic RT plans with a dose of 45.0 Gy in 25 fractions in patients with GEJ adenocarcinoma. 

## 2. Methods

### 2.1. Study Population

GastroPET is an academic investigator-initiated prospective, multicenter, interventional, non-randomized phase II exploratory clinical trial evaluating FDG-PET scan as a biomarker of tumor metabolic response to the standard POC treatment of locally advanced GEJ adenocarcinoma. The trial was approved by the Institutional Ethics Committee of Masaryk Memorial Cancer Institute, protocol code 2017/2123/MOU, date of approval 25 July 2017.

Eligibility criteria included the biopsy-proven, locally advanced resectable adenocarcinoma or esophagogastric junction (Siewert I–III) stage Ib–IIIc. Eligible patients had to be fit for oxaliplatin-fluoropyrimidine-(docetaxel) containing chemotherapy (FOLFOX or FLOT), and tumors were deemed R0 resectable after consultation with the institutional multidisciplinary tumor board. Key exclusion criteria were age <18 years, Eastern Cooperative Oncology Group (ECOG) score >2, life expectancy <3 months, uncontrolled tumor bleeding, and previous chemotherapy, radiotherapy, or endoscopic therapy for early-stage cancer within the last 3 months. Before treatment, all enrolled patients underwent fiberoptic esophagogastroscopy, endoscopic ultrasound, and initial pretreatment PET/CT imaging. Baseline standard uptake values (SUV) were determined for the tumor and involved nodes. The initial PET/CT was then followed by the first cycle of the preoperative FLOT regimen.

After the first cycle of preoperative chemotherapy, an interim PET/CT scan was performed to evaluate metabolic response to guide further preoperative treatment. Patients with a decrease in the SUV mean >35% compared to the initial PET scan were considered to be metabolic responders and continued for two further cycles of preoperative FLOT chemotherapy. Patients with a decrease in the SUV mean <35% compared to the initial PET scan were deemed metabolic non-responders and were switched to concurrent chemoradiotherapy consisting of five times weekly carboplatin at the area under the concentration versus time curve 2 mg/mL/min and paclitaxel at 50 mg/m^2^, together with concurrent radiotherapy (45 Gray (Gy) in 25 fractions, 1.8 Gy per daily fraction, five days per week for five weeks with no additional boost). Patients from both arms of preoperative treatment further followed the original study protocol, which consisted of radical surgery and follow-up.

All patients from the non-responding arm with concurrent chemoradiotherapy were included in this secondary analysis consisting of a virtual planning study evaluating the role of an increased dose of PCRT. This planning study consisted of the subsequent virtual boost of a 9 Gy in 5 fractions, added to the originally applied RT plans with 45 Gy in 25 fractions. The non-responding arm was the only inclusion criterion for the virtual planning study.

### 2.2. Patients’ Characteristics

A total of 16 patients (pts) with adenocarcinoma of GEJ (12 men and 4 women) were deemed non-responders and enrolled in this planning study. The average age was 67 years with a median of 67 years (range 52–76). There were 7 pts with initial TNM (Tumor, Node, Metastasis) classification T3N0M0, 6 pts with T3N1M0, one with T4aN0M0, and two with T4aN1M0. According to the tumor histology (Laurén classification), there were 7 intestinal, 5 diffuse types of adenocarcinomas, and 4 without were adenocarcinomas without further specification. Grade 1 was in 3, grade 2 in 4, and grade 3 in 9 cases. All patients fulfilled the treatment plan according to the non-responding arm of the core study protocol. A total of 14 out of 16 patients underwent successive surgical treatment; R0 resection was reached in 12 cases, R1 resection in two cases. The Mandard´s tumor regression score (TRG) after preoperative treatment was assessed in 12 out of 16 pts. No TRG 0 and 1was found, three patients reached TRG 2, five TRG 3, and four TRG 4 with no case of TRG 5.

### 2.3. Radiotherapy 

The prescribed dose of concurrent RT in the preoperative setting was 45.0 Gy given in 25 fractions of 1.8 Gy on 5 days per week. In the presented in silico planning study, the prescribed dose was virtually increased by the additional boost to the primary tumor and involved nodes at 9 Gy in five fractions of 1.8 Gy to a total dose of 54.0 Gy. Before initiating radiation treatment planning and delivery, all patients underwent the fiberoptic esophagogastroscopy, endoscopic ultrasound, and pretreatment PET/CT (baseline and interim), as mentioned above. The radiation therapy planning process consisted of a standard CT in the supine position with the intravenous administration of contrast medium. CT slice-thickness was not larger than 3 mm. The median number of CT slices was 123 (range 105–144). An illustration of the pre- and post-radiotherapy CT images for one selected patient is shown in the Appendix A as Appendix A.

Several target volumes were defined as follows: gross tumor volume (GTV_tumor) included the tumor site and its extent defined by FDG-PET—computed tomography (hybrid PET/CT scans), so the areas with SUV with all available pretreatment examinations such as fiberoptic endoscopy, endoscopic ultrasound (EUS) were assessed as well. GTV was also determined for the involved lymph nodes (GTV_nodal), including all visible CT lymphnodes along the GEJ and lesser curvature, and PET/CT active lymphnodes in another lymph node stations, described further as elective for in clinical target volume (CTV). CTV consisted of the sum of all additional clinical target volumes as follows: CTV_tumor was created by adding a margin of 1–1.5 cm radially and 3 cm cranially and 3–5 cm distally to GTV_tumor to include the position of the tumor after the registration of planning CT and initial PET/CT on GEJ to cover possible variations in the shape of the upper part of stomach on these initial examinations. CTV_nodal was created by adding 0.5 cm to GTV_nodal. CTV was further enlarged to encompass the elective lymph node stations –paraoesophageal, paracardial, perigastric—along the lesser curvature, short gastric vessels, splenic artery, splenic hilum, coeliac axis, or an entire proximal third of the stomach in case of the tumor with suspicious spread within the 5 cm from the gastroesophageal junction. An additional margin of 0.8–1 cm was created to define the planning target volume (PTV) to correct daily setup variation and organ motion. Additional virtual boost volume consisted of adding 1 cm to GTV_tumour and 0.5 cm to GTV_nodal in all directions creating CTV_boost. For the boost volume, the lymphnodes were initially deemed PET/CT-positive only from stations paraesophageal, along GEJ, lesser curvature or coeliac axis were included. The lymphonodes from the other more distant lymphonode stations were excluded from the boost volume even if initially deemed PET/CT-positive, but no such case appeared in this planning study. Another 0.8 cm was added to define the PTV_boost volume. An illustration of the treatment planning difference for one selected patient is shown in the Appendix A as Appendix A.

The following organs at risk (OARs) were contoured: whole organs—heart, lungs, kidneys, liver, bowels small and large intestine loops separately, and whole peritoneal cavity excluding CTV contoured two slices below PTV. All treatment volumes were contoured and double-checked by experienced radiation oncologists (M.S., P.B., and T.K.). Only volumetric modulated RT techniques (VMAT) were used. The Eclipse Planning Software, version 15.6, with AAA algorithm (Varian, Palo Alto, CA, USA) was used to generate the treatment plans. A single-phase coplanar VMAT plan was calculated on the planning CT scan, tailored to achieve optimal PTV coverage while respecting the dose volume constraints. The plan was typically delivered from 1 or 2 volumetric modulated arcs with the gantry angles in the ranges 0–360°. The exact gantry angle range was not mandated and was adjusted to meet the optimal coverage of PTVs and dose volume constraints. Only 10 megavoltage photon energy was used. The same planning process was utilized for additional boost and the summary plan was then assessed. 

Dose prescription and recordings were in accordance with recommendations of the International Commission on Radiation Units and Measurements (ICRU) 50/62 and 83. The dose homogeneity within the planning volume was within −5% and +7% of the prescribed dose. The PTV should be encompassed by the 95% isodose-volume. Underdosage was only allowed if requested by the proximity of serial OAR. Doses on OARs complied with the Quantitative Analysis of Normal Tissue Effects in the Clinic (QUANTEC) recommendations [14]. 

### 2.4. Treatment Plan Evaluation and NTCP Modeling

The treatment plans were then analyzed according to dose–volume histograms (DVHs) data. Following DVH parameters were evaluated, and the differences were set between a primary plan (D 45 Gy) and the virtual plan with an additional boost of 9.0 Gy (D 54 Gy): the mean lung, heart, and liver doses, median left and right kidney doses, the volume of the isodose of 25 Gy (V_25Gy_ isodose) for each lung, bilateral lung (sum of both lungs) volume of the isodose of 20 Gy (V_20Gy_ isodose), the heart volume of 30 Gy isodose (V_30Gy_ isodose), the small bowel volume of 15 Gy isodose (V_15Gy_ isodose), the liver volume of 35 Gy isodose (V_35Gy_ isodose), the volumes of the isodoses of 15 and 45 Gy (V_15Gy_ and V_45Gy_ isodose) for peritoneal space, and the maximal dose on the spinal cord. OAR constraints of DVH parameters are shown in Table 1.

Normal tissue complication probability (NTCP) for heart, spine, lungs, kidneys, liver, peritoneal cavity, and small bowel was calculated for basic dose 25 × 1.8 Gy, and escalated boost 25 × 1.8 Gy + 5 × 1.8 Gy. Lyman-Kutcher-Burman (LKB) model was employed, describing the sigmoidal dose-response curve of normal tissues at the software BioGrayPlus, (East Slovakia Oncology Institute, Kosice, Slovakia) Version 2.0.3.1104 [15]. This software uses model parameters based on QUANTEC project for these G3 toxicity endpoints: Kidney—Clinical Nephritis; Heart—Pericarditis and pancarditis; Spine—Myelitis, necrosis; Liver—failure; Lungs—pneumonitis; Small bowel—obstruction, perforation; Peritoneal cavity—obstruction, perforation. The parameters *n*—volume dependence; NTD50(1)—dose sensitivity; *m*—slope of DVH; reference volumes for *v* = 1 (whole organ) are summarized in Table 2.

### 2.5. Statistical Analysis

Primary plans with the prescribed dose of 45 Gy/25 fractions were compared with particular summary plans consisting of primary plans of 45 Gy/25 fractions with the additional virtual boost of 9.0 Gy in 5 fractions. Differences for each defined parameter were determined between particular fractionation schemes. The DVH parameters and the differences were described using standard summary statistics, i.e., median and range. Moreover, mean DVH was estimated for PTV and specific OAR. The coverage of individual PTVs by 95% isodose in primary and boost plans was expressed by the value of the Van’t Riet conformity index (CI), with the most optimal value being equal to 1, which means a practically unachievable situation when the 95% isodose exactly fits the defined PTVs. To compare the NTCP values between groups, a two-tailed paired Wilcoxon test was used with a common significance level of 0.05. All statistical analyses were performed employing R version 4.0.3.

## 3. Results

### Radiotherapy Plans Evaluation

All radiation treatment plans met the study criteria regarding the dose homogeneity within the PTVs in the primary and the boost plans. Additionally, the matching the shape of the PTVs by the 95% isodose was adequate—average Van´t Riet conformity index for the primary treatment was 0.89 with the median value of 0.90 (range 0.84–0.94), and for the boost plans average 0.87 with a median of 0.85 (range 0.83–0.97). The extracted dose–volume characteristics of all OARs are summarized in Table 3.

The DVH parameters for the lung sum (mean and V_20Gy_ isodose), kidney (median), spinal cord (maximum), the small bowel (V_15Gy_ isodose) and liver (V_35Gy_ isodose) did not exceed the limits in all the cases for primary and D 54 Gy plans.

The mean heart dose exceeded the limit of 26 Gy in two cases in D 54 Gy plan. Peritoneal space volume criterion V_45Gy_ < 195 ccm was not maintained within evaluation limits in three cases in D 54 Gy plan, while the parameter V_15Gy_ isodose was exceeded the limit of 825 ccm in one case in both fractionation schedules.

The assessed volume of 25 Gy isodose exceeded the limit of 10% in most cases for the left lung in both dose schedules. For the right lung, one case of D 45 Gy plan and three cases of D 54 Gy plan were under the limit of 10%. 

DVH means of PTV and OARs for both plans are shown in Figure 1. A specific comparison of DVH means between plans is illustrated in Figure 2 for PTV, heart, lung sum and peritoneal space. Individual DVH for each patient are included in the Appendix A as Appendix A.

Calculated values of NTCP are summarized in Table 4. 

The NTCP values for heart, spine, liver and kidneys were zero or near zero for both fractionation schemes. The values for lungs were more variable, according to the distance of PTV from the lung. Dose escalation results only in a low increase in median NTCP value (median 3.15% vs.4.05 %, for 25 × 1.8 Gy and 25 + 5 × 1.8 Gy, respectively, *p* = 0.013). Median NTCP values for small or large intestines were also zero or near-zero. The potential movement of the intestine was not taken into consideration. When we used the same model for the whole peritoneal cavity, which is the peritoneal space, where the intestine can take place with certain probability, the value is much higher. In this case, the increment of NTCP for obstruction/perforation is more interesting than the NTCP value itself. Median values for 25 × 1.8 Gy, 25 + 5 × 1.8 Gy were 3.3%, and 14.25 %, respectively, *p* < 0.001.

## 4. Discussion

This planning study objectified the role of PET/CT-based dose increase on the tumor and involved nodes in the preoperative treatment of gastroesophageal adenocarcinoma. The use of PET/CT is also incorporated in the newest study dealing with GEJ tumors and has the potential to estimate the extent of disease with better accuracy [7]. While the dose on the tumor and involved nodes was significantly higher, the dose burden on OARs was exceeded in particular lungs and peritoneal space parameters. The lung parameter, where the volume of 25 Gy isodose should exceed 10%, was derived from breast cancer RT constraints and was used as an illustrative parameter although it was not validated for gastrointestinal RT. We only used it to show the dose burden on the lungs with lower doses. Otherwise, the lung dose constraints were not exceeded. The largest differences in the dose–volume parameter we found in the cases where the 45 Gy isodose should not overlap the volume of 195 ccm of peritoneal space [14]. This criterium was not met in three patients with a higher dose schedule. Despite its importance, it was set for 3D conformal radiotherapy planning [16].

The other constraints derived from protocols of neoadjuvant chemoradiation for rectal cancer. The risk of grade 3 small bowel toxicity less than 10% in cases if parameters V_15Gy_ < 275 ccm for individual loops and V_15Gy_ < 825 ccm for the peritoneal cavity were met. In our cases, the last of the mentioned constraints was slightly exceeded in only one case. Nevertheless, it is appropriate to report all these parameters to assess the potential risk of gastrointestinal toxicity, and in some cases, it may be improved by additional plan optimization. The NTCP modeling is a reliable method, but the model we used for the peritoneal cavity is assigned for the small intestine only. In the case of small intestine loops, no elevated risk of G3 complications was shown, but it must be interpreted with caution as DVH parameters for the peritoneal cavity seem to be more significant in this case. Our results in dose burden of risk organs are similar to the results of the planning study published by Li et al. [17] compared with the standardly planned VMAT with pinnacle auto-planning in lower esophageal cancer patients. The mean lung dose in our study (8.8 Gy and 10.0 Gy for dose primary and boost schedules, respectively) was comparable to automated planning (9.83 Gy) and better than standard VMAT plans (11.9 Gy). In lung sum, V_20Gy_ isodose was also comparable, or even better, in our case (22.3% and 26.3% vs. 13.9% and 15.7%).

Our parameters were worse for heart V_30Gy_ (auto/manual planning 13.6 ccm/17.1 ccm vs. 22.9 ccm and 27.1 ccm for our two dose schedules). In fact, we allowed higher doses to the heart to spare the lungs to minimize the surgical complications considering a relatively worse oncological prognosis of these patients, where the intention to avoid possible serious perioperative complications is more important than reducing the risk of late ischemia. On the other hand, the doses causing an increased risk of pericarditis remained within the limits (V_30Gy_ < 46%). The mean doses on the liver in the published planning study were worse in our study (7.8% and 10.4% vs. 18.7% and 20.5%). The worse mean doses may be due to lower primary tumor placement focusing on better lung and peritoneal space sparing than in the cited study. The liver dose volume parameter V_35Gy_ values are comparable to V_30Gy_ in the cited study (9.60% for automated and 12.4% for standard VMAT planning). 

In the study on 20 patients, comparing manual and hybrid automated (script-based planning and knowledge-based planning combination) treatment planning, similar results were shown regarding dose burden on the heart, lungs and liver [18]. However, there are limitations in the direct comparison of the cited studies with our cohort. Both studies used 60 Gy resp. 61.4 Gy to the tumor, and the locations of the tumors were somewhat different than those in our study: they were mostly located above the diaphragm, and despite sometimes considerable PTVs, the authors did not have to deal with peritoneal space-sparing. 

The importance of this planning study is related to the observation that achieving a significant TRG and pT0 resection could be associated with better treatment outcomes, and even have an impact on overall survival [9]. The most common degree of regression in our study, after primary treatment, was Mandard TRG 3 (not yet published) or lower, which was shown to be associated with worse overall treatment results [10]. Based on this observation, the applied dose of 45 Gy seems to not be sufficient. Therefore, the potential RT dose increase might improve the treatment results. In addition, reaching R0 resection is fundamental for the long-term survival of these patients, and a higher rate of R0 resection was associated with neoadjuvant (chemo)radiation treatment [11,19]. Of course, this treatment is associated with non-negligible toxicity [19] and it is very difficult to estimate the extent of the potential adverse events and safety of the surgical procedure after such treatment augmentation. In the core trial, there are indications that there was no statistically significant difference in overall G3 toxicity between the neoadjuvant chemotherapy (18%) and neoadjuvant concomitant chemoradiotherapy (11%) group (*p* = 0.685, not yet published). This fact is potentially encouraging and favors the possibility of cautious dose escalation.

Considering the importance of reaching R0 resections, a recent meta-analysis with more than 13,000 patients reached the opposite conclusions [2]. Although a higher grade of R0 resections was present, no survival advantage was demonstrated after incorporating RT into the preoperative treatment of GEJ [2]. This metaanalysis has several limitations. It was a retrospective analysis of prospectively collected data, with several cofounding factors inherent to large dataset analysis, and it also lacks important pieces of information. In addition to information on chemotherapy, it also lacks detailed information on the radiation technique used, and because the data are derived from the period of 2004–2015, older and simpler RT techniques were likely utilized. On the other hand, the results of the prospective CROSS trial demonstrated, in addition to a higher rate of R0 resections, a longer overall survival and, in the last update, also suggested a reduced incidence of distant metastases [11]. This evidence, together with the fact that reaching complete pathological remission, demonstrated encouraging treatment results, leading to long-term survival [11,20]. This has led us to the idea of a potential dose increase with a higher probability of tumor control. Although our results showed acceptable doses for OARs with the implementation of a modern RT technique, this is still a hypothesis-generating planning study, which serves as a crucial prerequisite of prospective trials focused on the safety and efficacy of dose escalations using modern RT techniques.

## 5. Conclusions

With new state-of-the-art radiation treatment, we demonstrated the ability to relatively safely increase the dose for tumor and involved dose in the preoperative setting with an acceptable dose volume burden on selected OARs in the adenocarcinoma of GEJ. This planning study might be interpreted as a solid basis for future studies dealing with RT in this field.

## Figures and Tables

**Figure 1 medicina-57-01334-f001:**
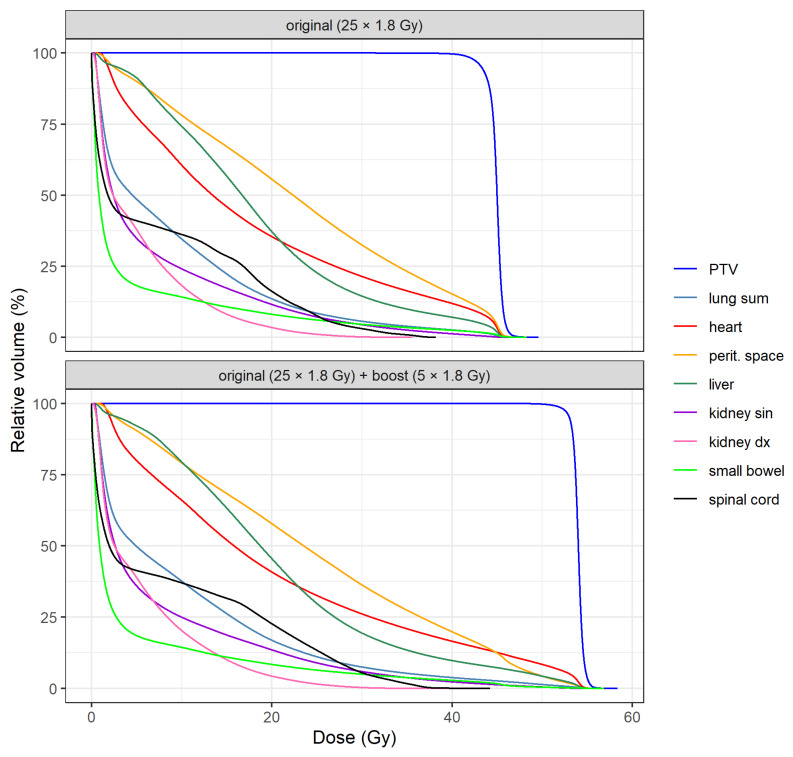
Dose-volume histogram means of the planning target volume (PTV) and organs at risk (OARs) for primary (top) and boost (bottom) plans.

**Figure 2 medicina-57-01334-f002:**
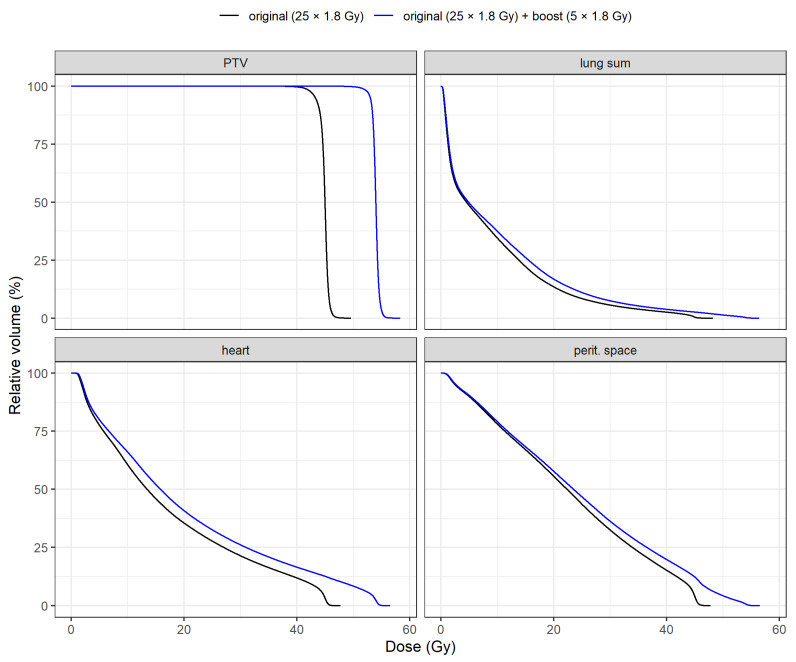
Comparison of dose-volume histogram means means between primary (black) and boost (blue) plans.

**Table 1 medicina-57-01334-t001:** OAR constraints of dose-volume histogram parameters.

OAR	Parameter	Limit
lung sum	mean	20 Gy
	V_20Gy_	35%
lung sin/dx	V_25Gy_	10%
heart	mean	26 Gy
	V_30Gy_	46%
small bowel	V_15Gy_	275 ccm
peritoneal space	V_45Gy_	195 ccm
	V_15Gy_	825 ccm
liver	mean	20 Gy
	V_35Gy_	66%
kidney sin/dx	median	15 Gy
spinal cord	max	45 Gy

Abbreviations: Gy—Gray, OAR—organ at risk.

**Table 2 medicina-57-01334-t002:** Parameters of Lyman-Kutcher-Burman NTCP model based on QUANTEC project used by BioGrayPlus software.

Parameter	Organ						
	Kidneys	Heart	Spine	Liver	Lungs	Small Bowel	Peritoneal Cavity
*n*	0.7	0.64	0.05	0.69	1	0.15	0.15
*m*	0.1	0.13	0.18	0.15	0.39	0.16	0.16
NTD50	32.3	50.6	71.6	45	31.4	58	58
α/β	3.25	2	2	1.5	3.7	7	7

Description: *n*—volume dependence; NTD50(1)—dose sensitivity; *m*—slope of DVH; reference volume for *v* = 1 (whole organ).

**Table 3 medicina-57-01334-t003:** Dose–volume histogram parameters.

		D 45 Gy	D 54 Gy	Difference
lung sum (mean, Gy)	median	8.8	10.0	1.1
	range	5.2–14.9	6.60–17.29	0.6–2.4
	limit exceeded	0 (0%)	0 (0%)	
lung sum (V_20Gy_, %)	median	13.9	15.7	2.7
	range	5.0–24.1	6.4-34.4	0.7–10.3
	limit exceeded	0 (0%)	0 (0%)	
lung sin (V_25Gy_, %)	median	16.3	19.5	2.8
	range	4.3–25.8	6.0–27.9	0.3–8.7
	limit exceeded	11 (75%)	12 (80%)	
lung dx (V_25Gy_, %)	median	5.1	5.8	1.8
	range	0.4–11.2	0.9–16.5	0.4–5.3
	limit exceeded	1 (7%)	3 (20%)	
heart (mean, Gy)	median	17.2	20.0	2.5
	range	14.3–29.5	16.5–33.8	2.1–4.3
	limit exceeded	1 (6%)	2 (13%)	
heart (V_30Gy_, %)	median	22.9	27.1	4.8
	range	11.7–53.3	15.0–60.5	2.7–8.3
	limit exceeded	1 (6%)	1 (6%)	
small bowel (V_15Gy_, ccm)	median	87.7	93.8	1.3
	range	11.2–181	12.4–183	0.2–6.1
	limit exceeded	0 (0%)	0 (0%)	
perit. space (V_45Gy_, ccm)	median	24.6	80.7	66.8
	range	0.1–168.7	0.2–278	0.1–164
	limit exceeded	0 (0%)	3 (19%)	
perit. space (V_15Gy_, ccm)	median	561.7	572.3	11.1
	range	134–870	134–880	0.75–72.7
	limit exceeded	1 (6%)	1 (6%)	
liver (mean, Gy)	median	18.7	20.5	2.60
	range	12.5–24.4	15.0–28.0	1.0–3.6
	limit exceeded	7 (44%)	9 (56%)	
liver (V_35Gy_, %)	median	10.4	13.8	3.2
	range	5.4–20.1	7.7–23.9	2.02–7.3
	limit exceeded	0 (0%)	0 (0%)	
kidney sin (median, Gy)	median	2.3	2.5	0.2
	range	1.0–13.0	1.1–14.8	0.1–1.8
	limit exceeded	0 (0%)	0 (0%)	
kidney dx (median, Gy)	median	2.3	2.4	0.1
	range	0.5–12.7	0.6–15.3	0.1–2.6
	limit exceeded	0 (0%)	1 (6%)	
spinal cord (max, Gy)	median	29.3	32.7	4.5
	range	16.7–37.4	21.5–43.5	1.5–7.4
	limit exceeded	0 (0%)	0 (0%)	

Description: D 45 Gy—primary plan; D 54 Gy—plan with an additional boost of 9 Gy. Abbreviations: Gy—Gray, OAR—organ at risk.

**Table 4 medicina-57-01334-t004:** Calculated NTCP values.

Median NTCP for Organs at Risk (%)								
Dose (Gy)	Organ							
	Kidney Right	Kidney Left	Heart	Spine	Liver	Lungs	Small Bowel	Peritoneal Cavity
25 × 1.8	0	0	0	0	0	3.15	0	3.3
25 + 5 × 1.8	0	0	0	0	0	4.05	0	14.25

## Data Availability

The data presented in this study are available on request from the corresponding author.

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
