# Peer review of "Preoperative Chemoradiotherapy for Gastroesophageal Junction Adenocarcinoma Modified by PET/CT: Results of Virtual Planning Study"

_medicina, 2021, doi:10.3390/medicina57121334_

Round 1

Reviewer 1 Report

I consider the revised version of this resubmitted manuscript has been improved and adresses to my previous Comments. The subject of the article is very interesting. As a minus, there are only 16 patients included in the study, and this gives limitted statistical value to the results. However, the paper is well documented and presents a novel approach to the radiotherapy of the gastroesofageal junction cancers and, în my opinion, it is worth to be published 

Author Response

Thank you for your review and comments. We agree that the sample size is not large. This fact is mainly influenced by the recruitment to the original study.

Reviewer 2 Report

This is a secondary analysis of the data obtained from an investigator-initiated phase II trial of PET/CT-guided neoadjuvant therapy of locally advanced adenocarcinoma of the gastro-oesophageal junction. Patients considered as non-responders to chemotherapy based on less than 35% decline in lesion SUVmean assessed on interim FDG PET/CT compared with baseline FDG PET/CT underwent radiotherapy – 45 Gy in 25 fractions. For the current study, the authors modeled dose boosting by 9 Gy to 54 Gy administered to the primary tumor and PET-positive nodes.

The study has some merits and may contribute to the growing literature on this subject. The study showed the feasibility of virtual dose boosting, a practice that may be helpful in guiding radiotherapy planning with higher doses in the current era of advanced RT techniques.

Specific comments  

  1. Introduction: “…and the better local control and longer follow up showed reduced locoregional recurrences in the treatment arm, and to a lesser extent reduced systemic recurrences [6].” Treatments were administered to patients in both arms of the trial. Please specify which arm is referred to as “treatment arm” in this highlighted clause.
  2. Abbreviations: Please define all abbreviations at first use. For example, abbreviations such as R0, GC, pT0 were not defined at first use in the main text.
  3. Introduction: “..and the fact that higher dose usually means the higher probability to reach the complete remission of the disease led us to conduct this planning study evaluating the safety of possible boosting the primary tumor and involved nodes in GEJ cancers..” Please provide references to published studies to support this assumption.
  4. Introduction: Provide a brief explanation on virtual dose boosting to provide potential readers with the background knowledge to understand what was done in this study.
  5. Introduction: Please report the aim of the study using the past tense.
  6. Methods: This section is very difficult to understand as it is not organized in a manner that gives a logical flow to what was done in the study. For example, after describing the primary trial in the first paragraph of the Methods, the authors began the second paragraph with this sentence: “After the first cycle of preoperative chemotherapy, a repeated PET scan was performed to evaluate metabolic response to guide further preoperative treatment.” This sentence leaves the reader wondering what happened before the first cycle of chemotherapy. I suggest the authors re-organize this section starting with the description of the study and a description of how patients were recruited into the study (including inclusion and exclusion criteria). The rest of the Methods should be organized in a manner that will allow potential readers to follow through on what was done in the study. Please describe the baseline PET/CT as well as interim PET/CT for response assessment.
  7. Statistical analysis: Please include the information regarding the statistical package used for data analysis.
  8. Discussion – lines 311-312: Please specify the organs in which dose burden was exceeded.

Round 2

Reviewer 2 Report

Thank you for revising your manuscript. The quality of the manuscript, especially the methods, is now much improved. I have no further comments.